# Unsupervised Brain Anomaly Detection and Segmentation with Transformers

**Walter Hugo Lopez Pinaya**[1]         WALTER.DIAZ_SANZ@KCL.AC.UK
**Petru-Daniel Tudosiu**[1]         PETRU.TUDOSIU@KCL.AC.UK
**Robert Gray**[2]         R.GRAY@UCL.AC.UK
**Geraint Rees**[3]         G.REES@UCL.AC.UK
**Parashkev Nachev**[2]         P.NACHEV@UCL.AC.UK
**Sébastien Ourselin**[1]         SEBASTIEN.OURSELIN@KCL.AC.UK
**M. Jorge Cardoso**[1]         M.JORGE.CARDOSO@KCL.AC.UK

[1] *Department of Biomedical Engineering, School of Biomedical Engineering & Imaging Sciences, King's College London, London, UK*

[2] *Institute of Neurology, University College London, London, UK*

[3] *Institute of Cognitive Neuroscience, University College London, London, UK*

## Abstract

Pathological brain appearances may be so heterogeneous as to be intelligible only as anomalies, defined by their deviation from normality rather than any specific pathological characteristic. Amongst the hardest tasks in medical imaging, detecting such anomalies requires models of the normal brain that combine compactness with the expressivity of the complex, long-range interactions that characterise its structural organisation. These are requirements transformers have arguably greater potential to satisfy than other current candidate architectures, but their application has been inhibited by their demands on data and computational resource. Here we combine the latent representation of vector quantised variational autoencoders with an ensemble of autoregressive transformers to enable unsupervised anomaly detection and segmentation defined by deviation from healthy brain imaging data, achievable at low computational cost, within relative modest data regimes. We compare our method to current state-of-the-art approaches across a series of experiments involving synthetic and real pathological lesions. On real lesions, we train our models on 15,000 radiologically normal participants from UK Biobank, and evaluate performance on four different brain MR datasets with small vessel disease, demyelinating lesions, and tumours. We demonstrate superior anomaly detection performance both image-wise and pixel-wise, achievable without post-processing. These results draw attention to the potential of transformers in this most challenging of imaging tasks.

**Keywords:** Transformer, Unsupervised Anomaly Segmentation, Anomaly Detection, Neuroimaging, Vector Quantized Variational Autoencoder

## 1. Introduction

Transformers have revolutionised language modelling, becoming the de-facto network architecture for language tasks (Radford et al., 2018, 2019; Vaswani et al., 2017). They rely on attention mechanisms to capture the sequential nature of an input sequence, dispensing with recurrence and convolutions entirely. This mechanism allows the modelling of dependencies of the inputs without regard to their distance, enabling the acquisition of complex long-range relationships. Since the approach generalises to any sequentially organised data,

applications in other areas such as computer vision are increasingly seen, with impressive results in image classification (Chen et al., 2020a; Dosovitskiy et al., 2020) and image synthesis (Child et al., 2019; Esser et al., 2020; Jun et al., 2020). The power to absorb relationships varying widely in their distance makes transformers of potential value in the arguably the hardest of neuroimaging tasks: anomaly detection.

The detection and segmentation of lesions in neuroimaging support an array of clinical tasks, including diagnosis, prognosis, treatment selection and mechanistic inference. However, the fine characterisation of these lesions requires an accurate segmentation which is generally both ill-defined and dependent on human expertise (Kamnitsas et al., 2017). Manual segmentation is expensive and time-consuming to obtain, greatly limiting clinical application, and the scale and inclusivity of available labelled data. Qualitative, informal descriptions or reduced measurements are often used instead in clinical routine (Porz et al., 2014; Yuh et al., 2012). For this reason, the development of accurate computer-aided automatic segmentation methods has become a major endeavour in medical image research (Menze et al., 2014). Most methods, however, depend on an explicitly defined target class, and are sensitive to the scale and quality of available labelled data, a sensitivity amplified by the many sources of complex variability encountered in clinical neuroimaging. Under real-world distributional shift, such models behave unpredictably, limiting clinical utility.

In recent years, many machine learning algorithms have been proposed for automatic anomaly detection. To overcome the necessity of expensive labelled data, unsupervised methods have emerged as promising tools to detect arbitrary pathologies (Baur et al., 2018, 2020b; Chen et al., 2020b; Pawlowski et al., 2018; Schlegl et al., 2017), relying mainly on deep generative models of normal data to derive a probability density estimate of the input data defined by the landscape of normality. Pathological features then register as deviations from normality, avoiding the necessity for either labels or anomalous examples in training. The state of the art is currently held by variational autoencoder (VAE)-based methods (Baur et al., 2020a) which try to reconstruct a test image as the nearest sample on the learnt normal manifold, using the reconstruction error to quantify the degree and spatial distribution of any anomaly. This approach's success is limited by the fidelity of reconstructions from most VAE architectures (Dumoulin et al., 2016), and unwanted reconstructions of pathological features not present in the training data, suggesting a failure of the model to internalise complex relationships between remote imaging features.

In an effort to address these problems, we propose a method for unsupervised anomaly detection and segmentation using transformers, where we learn the distribution of brain imaging data with an ensemble of Performers (Choromanski et al., 2020). We create and evaluate a robust method and compare its performance on synthetic and real datasets with recent state-of-the-art unsupervised methods.

## 2. Background

The core of the proposed anomaly detector is a highly expressive transformer that learns the probability density function of 2D images of healthy brains. This requires us to express each image's contents as a sequence of observations on which transformers-like models can operate. Owing to the size and complexity of brain imaging data, instead of learning the distributions on individual pixels directly, we use the compact latent discrete representation

of a vector quantised variational autoencoder (VQ-VAE) (Razavi et al., 2019; van den Oord et al., 2017). This approach allows us to compress the input data into a spatially smaller quantised latent image, thus reducing the computational requirements and sequence length, making transformers feasible in neuroimaging applications.

## 2.1. Vector quantized variational autoencoder

The VQ-VAE (Razavi et al., 2019; van den Oord et al., 2017) is a model that learns discrete representations of images. It comprises an encoder $E$ that maps observations $x \in \mathbb{R}^{H \times W}$ onto a latent embedding space $\hat{z} \in \mathbb{R}^{h \times w \times n_z}$, where $n_z$ is the dimensionality of each latent embedding vector. Then, an element-wise quantization is performed for each spatial code $\hat{z}_{ij} \in \mathbb{R}^{n_z}$ onto its nearest vector $e_k$ , $k \in 1, ...K$ from a codebook, where $K$ denotes the vocabulary size of the codebook. This codebook is learnt jointly with the other model parameters. A decoder $G$ reconstructs the observations $\hat{x} \in \mathbb{R}^{H \times W}$ from the quantized latent space. We obtain the latent discrete representation $z_q \in \mathbb{R}^{h \times w}$ by replacing each code by its index $k$ from the codebook.

## 2.2. Transformers

After training the VQ-VAE, we can learn the probability density function of the latent representation of healthy brain images using an autoregressive model. In recent studies, the transformer-based approaches have consistently outperformed other autoregressive models (Esser et al., 2020). The defining characteristic of a transformer is that it relies on attention mechanisms to capture the interactions between inputs, regardless of their relative position to one another. Each layer of the transformer consists of a (self-)attention mechanism described by mapping an intermediate representation with three vectors: query, key, and value vectors. Since the output of this attention mechanism relies on the inner products between all elements in the sequence, its computational costs scale quadratically with the sequence length. Several "efficient transformers" have recently been proposed to reduce this computational requirement (Tay et al., 2020). Our study uses the Performer, a model which uses an efficient (linear) generalized attention framework implemented by the FAVOR+ algorithm (Choromanski et al., 2020). This framework provides a scalable estimation of attention mechanisms expressed by random feature map decompositions, making transformers feasible for longer sequences, of the size needed for neuroimaging data.

To model brain images, after assuming an arbitrary ordering to transform the latent discrete variables $z_q$ into a 1D sequence $s$, we can train the transformer to maximize the training data's log-likelihood in an autoregressive fashion - similar to training performed on language modelling task. This way, the transformer learns the distribution of codebook indices for a position $i$ given all previous values $p(s_i) = p(s_i|s_{<i})$.

## 3. Proposed Method

### 3.1. Anomaly Segmentation

To segment an anomaly in a previously unseen, test image, first, we obtain the latent discrete representation $z_q$ from the VQ-VAE model. Next, we reshape $z_q$ into a sequence $s$, and we use the autoregressive transformer to obtain the likelihood of each latent variable value $p(s_i)$.

These likelihood values indicate which latent variable has a low probability of occurring in normal data. Using an arbitrary threshold (we empirically determined a threshold of 0.005 on a holdout set), we then can select indices with the lowest likelihood values and create a "resampling mask" indicating which latent variables are abnormal and should be corrected to produce a "healed" version of the image. We then replace these abnormal values with values sampled by the transformer. This approach attenuates the influence of the anomalies by replacing them by values that conform to the healthy distribution in the discrete latent space. This in-painted latent space can then reconstruct $\hat{x}'$ without the anomalies, in "healed" form. Finally, we obtain the pixel-wise residuals from the difference $|x - \hat{x}'|$. The anomaly in the new sample is finally segmented by thresholding the highest residuals values.

### 3.2. Spatial information from the latent space

Autoencoders are known for creating blurry reconstructions (Dumoulin et al., 2016). This can cause residuals with high values in areas of the image with high frequencies. To avoid these areas being mislabelled as anomalous, we used the spatial information present in the in-painted "resampling mask" of the latent space.

The resampling mask indicates the spatial location of the latent values with anomalies according to the transformer model. Since our VQ-VAE is relatively shallow, the latent space preserves most of the spatial information of the input data. Based on this, we used the resampling mask to avoid mislabelling of high-frequency regions. First, we upscaled the resampling mask from the latent space resolution to the input data resolution. Next, we multiply the residuals with the mask. This approach cleans areas of the residuals that were not specified as anomalous by our transformer but where the reconstructions might be largely due to lack of VQ-VAE capacity.

### 3.3. Multiple views of the latent space through reordering

Recent studies have reported some limitations of likelihood models, such as our autoregressive model, in identifying out-of-distribution samples (Nalisnick et al., 2018). Inspired by Choi et al. (2018), we made the detection of the anomalies more robust by using an ensemble of models. Using the same VQ-VAE model, we trained an ensemble of autoregressive transformers. However, unlike Choi et al. (2018), each one of our transformers uses a different reordering of the 2D latent image to create a sequence. This compels each transformer to use a different context of the latent image when predicting the likelihood of an element.

In our study, we focused on the raster scan class ordering. We obtain different orderings by reflecting the image horizontally, vertically, and both ways at the same time. We also define our orderings in images rotated 90 degrees, generating 8 different orderings from a single latent representation. Each resampled latent representation is independently reconstructed, i.e. each model independently creates a residuals map. We use the mean residual to segment the anomalies.

### 3.4. Image-wise Anomaly Detection

So far, the proposed methodology has been focusing on segmenting abnormalities. However, transformers can also be used to perform image-wise anomaly detection, i.e. detecting if an

abnormality exists somewhere in the image. To do so, we use the likelihood predicted by the transformers. Like the segmentation approach, first, we obtain the 1D latent representation $s$. Then, we use the transformers to obtain the likelihood $p(s)$ of each latent variable. To obtain the log-likelihood image-wise, we compute $logp(x) = logp(s) = \sum_i logp(s_i)$. Finally, we combined the predicted log-likelihood of each transformer (per orientation/ordering) by computing the mean value.

## 4. Experiments and Results

### 4.1. Experiment #1 – Anomaly Segmentation on Synthetic Data

To assess anomaly segmentation performance on synthetic data, we utilized a subsample of the MedNIST dataset[1], where we used the images of the "HeadCT" category to train our models. Our test set comprised 100 images contaminated with sprites (Matthey et al., 2017), thus producing ground-truth abnormality masks. We measure the performance using the best achievable DICE-score ($\lceil DICE \rceil$), which constitutes a theoretical upper-bound to a model's segmentation performance and is obtained via a greedy search for the residual threshold which yields the highest DICE-score on the test set. We also obtained the area under the precision recall curve (AUPRC) as a sensible measure for segmentation performance under class imbalance. We compared our results against state-of-the-art autoencoder models based on the architectures proposed in the (Baur et al., 2020a) comparison study, assessed in the same manner. We also performed an ablation study of the proposed method demonstrating the values of each contribution.

Table 1: Methods on anomaly segmentation using the synthetic dataset. The performance is measured with best achievable DICE-score ($\lceil DICE \rceil$) and AUPRC on the test set.

| Method | $\lceil DICE \rceil$ | AUPRC |
|---|---|---|
| AE (Dense) (Baur et al., 2020a) | 0.213 | 0.129 |
| AE (Spatial) (Baur et al., 2020a) | 0.165 | 0.093 |
| VAE (Dense) (Baur et al., 2020a) | 0.533 | 0.464 |
| VQ-VAE (reconstruction-based; van den Oord et al., 2017) | 0.457 | 0.346 |
| VQ-VAE + Transformer (Ours) | 0.675 | 0.738 |
| VQ-VAE + Transformer + Masked Residuals (Ours) | 0.768 | 0.808 |
| VQ-VAE + Transformer + Masked Residuals + Different Orderings (Ours) | **0.895** | **0.956** |

As presented in Table 1, the models based only on autoencoders had a highest $\lceil DICE \rceil$ of 0.533 (VAE). We observed an improvement in performance when using the transformer to in-paint the latent space, changing the VQ-VAE only performance from 0.457 to 0.675. The spatial information in the resampling mask also contributed by attenuating the impact of the blurry reconstructions (Figure 1), achieving a 0.768 score. Finally, the variability of the generative models with different orderings gave another boost in performance (for 8 different raster ordering models $\lceil DICE \rceil$=0.895).

### 4.2. Experiment #2 – Image-wise Anomaly Detection on Synthetic Data

Next, we evaluated our method to detect anomalous (out-of-distribution - OOD) images, again on a synthetic setting. Using the same models trained from Experiment #1, we ob-

---

1. Available at https://github.com/Project-MONAI/tutorials

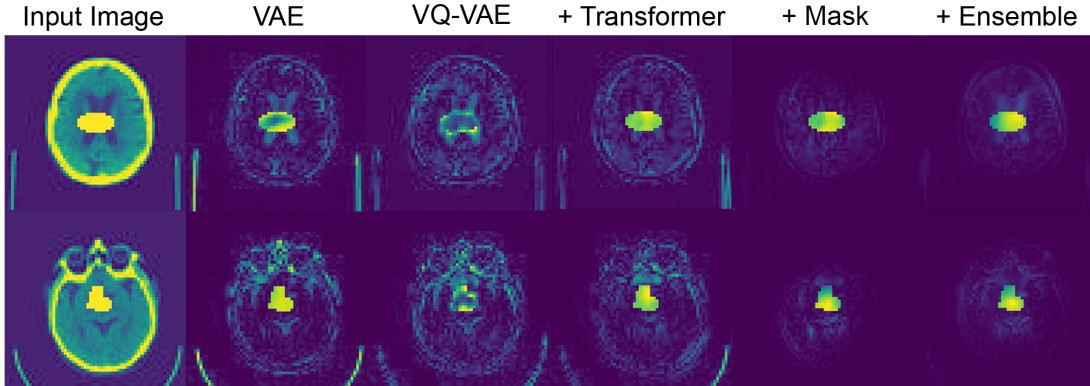

Figure 1: Residual maps on the synthetic dataset from the variational autoencoder and different steps of our approach.

tained the mean log-likelihood for each image. We used 1,000 images from the HeadCT class as the in-distribution test set, the 100 HeadCT images contaminated by sprite anomalies as the near OOD set, and 1.000 images of each other MedNIST classes as the far OOD set (see the appendix for details). We use the area under the receiver operating characteristic curve (AUROC) as performance metric, with in-distribution test set and out-of-distribution being the labels. This way, we have a threshold-independent evaluation metric. We also measure the AUPRC, where it provides a meaningful measure for detection performance in the presence of heavy class-imbalance. Finally, we also computed the false positive rate of anomalous examples when the true positive rate of in-distribution examples is at 95% (FPR95), 99% (FPR99) and 99.9% (FPR999).

Table 2: Performance of the methods on image-wise anomaly detection using the synthetic dataset.

|  | AUCROC | AUPRC In | AUPRC Out | FPR95 | FPR99 | FPR999 |
|---|---|---|---|---|---|---|
| vs. far OOD classes |  |  |  |  |  |  |
| VAE (Dense) (Baur et al., 2020a) | 0.298 | 0.855 | 0.060 | 0.986 | 0.996 | 0.996 |
| Our approach | **1.000** | **1.000** | **1.000** | **0.000** | 0.001 | 0.004 |
| Our approach with general purpose VQ-VAE | **1.000** | **1.000** | **1.000** | **0.000** | **0.000** | **0.000** |
| vs. near OOD classes |  |  |  |  |  |  |
| VAE (Dense) (Baur et al., 2020a) | 0.111 | 0.094 | 0.672 | 1.000 | 1.000 | 1.000 |
| Our approach | 0.921 | 0.988 | 0.707 | **0.409** | 0.885 | 0.885 |
| Our approach with general purpose VQ-VAE | **0.932** | **0.990** | **0.721** | 0.482 | **0.882** | **0.882** |

Table 2 shows that our transformer-based method achieved an AUCROC of 0.921 and 1.000 for near OOD and far OOD, respectively. This is a improvement compared to a method based on the error of reconstruction obtained from a VAE model. We also evaluated our method with a VQ-VAE trained to reconstruct all the categories from the MedNIST dataset ("general purpose VQ-VAE") and the ensemble of transformers trained on HeadCT images only. In this configuration, we try to mitigate the influence of the encoder in the anomaly detection. This approach would reduce the ability of the encoder to map an

OOD image to a familiar in-distribution latent representation, which could possibly affect the transformer performance. This new configuration achieves a slight better performance (AUCROC=0.932 for near OOD and AUCROC=1.000 for far OOD).

### 4.3. Experiment #3 – Anomaly Segmentation on Real Neuroimaging Data

To evaluate our method's performance on real world lesion data, we used the FLAIR images from the UK Biobank (UKB) dataset (Sudlow et al., 2015). We selected the 15,000 subjects, and their respective FLAIR images, with the lowest white matter hyperintensities volume, as provided by UKB, to train our models, as these subjects were the most radiologically normal. Then, we used 18,318 subjects from the remaining UKB dataset to evaluate our method to detect white matter hyperintensities (WMH).

In order to test for model generalisability, we also evaluated our method on three other datasets that also had FLAIR imaging data: the Multiple Sclerosis dataset from the University Hospital of Ljubljana (MSLUB) dataset (Lesjak et al., 2018), which contains multiple sclerosis lesions; the White Matter Hyperintensities Segmentation Challenge (WMH) dataset (Kuijf et al., 2019); and the Multimodal Brain Tumor Image Segmentation Benchmark (BRATS) dataset (Bakas et al., 2017, 2018; Menze et al., 2014) that contain tumours (more information about the datasets and pre-processing on appendix A).

Table 3: Results of the anomaly segmentation using real lesion data. We compared our models against the state-of-the-art autoencoder models based on the architecture proposed in Baur et al. (2020a). We measured the performance using the theoretically best possible DICE-score ($\lceil DICE \rceil$) and AUPRC on each dataset.

| UKB Dataset | $\lceil DICE \rceil$ | AUPRC |
|---|---|---|
| AE (Dense) (Baur et al., 2020a) | 0.016 | 0.005 |
| AE (Spatial) (Baur et al., 2020a) | 0.054 | 0.015 |
| VAE (Dense) (Baur et al., 2020a) | 0.016 | 0.006 |
| VQ-VAE (reconstruction-based; van den Oord et al., 2017) | 0.028 | 0.005 |
| VQ-VAE + Transformer + Masked Residuals + Different Orderings (Ours) | **0.232** | **0.159** |
| **MSLUB Dataset** | | |
| AE (Dense) (Baur et al., 2020a) | 0.041 | 0.016 |
| AE (Spatial) (Baur et al., 2020a) | 0.061 | 0.026 |
| VAE (Dense) (Baur et al., 2020a) | 0.039 | 0.016 |
| VQ-VAE (reconstruction-based; van den Oord et al., 2017) | 0.040 | 0.016 |
| VQ-VAE + Transformer + Masked Residuals + Different Orderings (Ours) | **0.378** | **0.272** |
| **BRATS Dataset** | | |
| AE (Dense) (Baur et al., 2020a) | 0.276 | 0.094 |
| AE (Spatial) (Baur et al., 2020a) | 0.531 | 0.215 |
| VAE (Dense) (Baur et al., 2020a) | 0.294 | 0.107 |
| VQ-VAE (reconstruction-based; van den Oord et al., 2017) | 0.331 | 0.125 |
| VQ-VAE + Transformer + Masked Residuals + Different Orderings (Ours) | **0.759** | **0.555** |
| **WMH Dataset** | | |
| AE (Dense) (Baur et al., 2020a) | 0.073 | 0.024 |
| AE (Spatial) (Baur et al., 2020a) | 0.150 | 0.054 |
| VAE (Dense) (Baur et al., 2020a) | 0.068 | 0.022 |
| VQ-VAE (reconstruction-based; van den Oord et al., 2017) | 0.100 | 0.036 |
| VQ-VAE + Transformer + Masked Residuals + Different Orderings (Ours) | **0.429** | **0.320** |

Our method showed a better performance than the autoencoder approaches from the literature in all datasets (Table 1 and Figure 2). Compared to the numbers in Baur et al.

(2020a), our autoencoder-based models got a lower performance on the common dataset (MSLUB dataset), where they achieved an DICE score of 0.271 with the AE (dense), 0.154 with the AE (spatial), and 0.323 with the VAE (dense). We believe that the discrepancy comes mostly from the significant post-processing of the Baur et al. (2020a) work as presented in Table 8 of this reference. Differences might also arise from the difference in resolution, as the DICE score is not invariant to resolution.

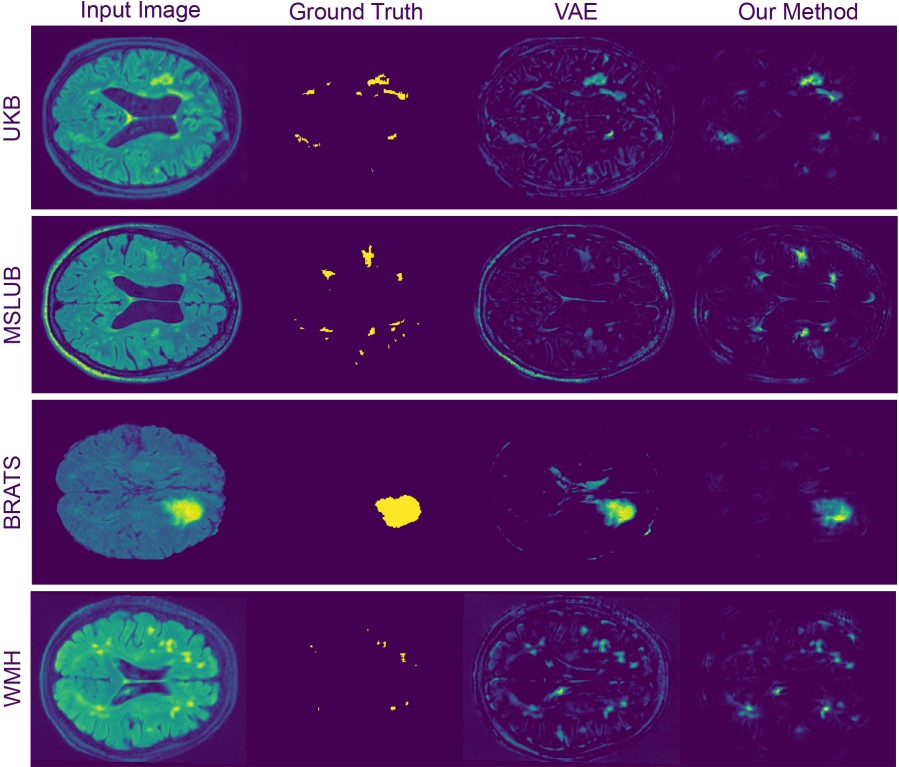

Figure 2: Residual maps on the real lesions from the variational autoencoder and our transformer-based method.

## 5. Conclusion

Automatically determining the presence of lesion and delineating their boundaries is essential to the introduction complex models of rich neuroimaging features in clinical care. In this study, we propose a novel transformer-based approach for anomaly detection and segmentation which achieves state-of-the-art results in all tested tasks when compared to competing methods. Transformers are making impressive gains in image analysis, and here we show that their use to identify anomalies holds great promise. We hope that our work will inspire further investigation of the properties of transformers for anomaly detection in medical images, the development of new network designs, exploration of a wider variety of conditioning information, and the application of transformers to other medical data.

## Acknowledgments

WHLP and MJC are supported by Wellcome Innovations [WT213038/Z/18/Z]. PTD is supported by the EPSRC Research Council, part of the EPSRC DTP, grant Ref: [EP/R513064/1]. PN is supported by Wellcome Innovations [WT213038/Z/18/Z] and the UCLH NIHR Biomedical Research Centre. This research has been conducted using the UK Biobank Resource (Project number: 58292).

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

## Appendix A. Experimental Details and Further Analysis

### A.1. Experiment #1 − Anomaly Segmentation on Synthetic Data

**Dataset**   To assess the performance on anomaly segmentation, we utilized a subsample of the MedNIST dataset, where we used the 2D images of the HeadCT category to train our VQ-VAE and transformer models (Figure A1). From the original 10,000 HeadCT images, we used 8,000 images as the training set and 1,000 images for the validation set. The test set was comprised of 100 images contaminated with sprites (i.e., synthetic anomalies) obtained from the dsprites dataset (Matthey et al., 2017). We selected the sprites images that overlapped a significant portion of the head, and their values were set as 0 or 1.

**Models**   Our VQ-VAE models had a similar architecture from van den Oord et al. (2017). The encoder consists of three strided convolutional layers with stride 2 and window size 4 × 4. All these convolution layers had a ReLU activation following them. This structure is followed by two residual 3×3 blocks (implemented as 3×3 conv, ReLU, 1×1 conv, ReLU).

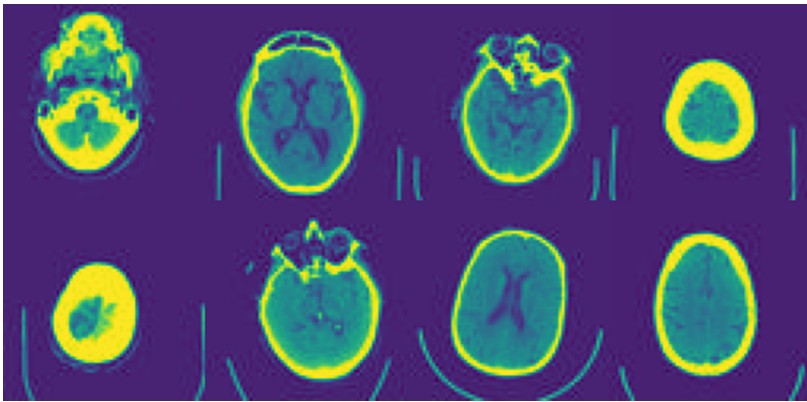

Figure A1: Examples of the training set (HeadCT class) from the MedNIST dataset.

The decoder similarly has two residual 3×3 blocks, followed by three transposed convolutions with stride 2 and window size 4×4. All the convolution layers have 256 hidden units. The inputted images have the dimension of 64x64 pixels which result in a latent representation of 8x8 latent variables. For this experiment, we used a codebook with 16 different codes. Our performers corresponded to the transformer's decoder structure with 24 layers with an embedding size of 256. The embedding, feed-forward and attention dropout had a probability of 0.1.

**State-of-the-art Models**   We compared our models against state-of-the-art autoencoder-based methods (AE dense, AE spatial, and VAE). In this experiment, we used an unified network architecture adapted from a recent comparison study (Baur et al., 2020a). Since MedNIST images are smaller than those used in the comparison study, our models did not have the first block (resolution of 64x64x32) and the last block (resolution of 128x128x32). All the other blocks and layers are similar to the original study. To train these models, we used the ADAM optimiser with learning rate 5e-4, an exponential learning rate decay with gamma of 0.9999, and we trained over 1,500 epochs with batch-size 256. Similar to Baur et al. (2020a), we used the held-out validation set to select the models to be assessed.

**Training settings**   To train the VQ-VAE models, we use the ADAM optimiser with learning rate 5e-4, an exponential learning rate decay with gamma of 0.999, and we trained over 1,500 epochs with batch-size 256. We train the codebook using an exponential moving average algorithm. To stabilise the codebook's learning, in the first 100 epoch, we warm up the moving average decay from a gamma decay of 0.5, gradually increasing to a gamma decay of 0.99. This allows the codes to adapt faster to the frequent changes at the beginning of the training. To train the performers, we used the ADAM optimiser with learning rate 5e-4, an exponential learning rate decay with gamma of 0.9999, and we trained over 1,500 epochs with batch-size 128. We used data augmentation to increase the number of training images. We randomly performed affine transformations (scale, translate, and rotate operations) and horizontally flip the images.

**Different Ordering Classes**   In our study, we also analysed 3 other classes of orderings (Figure A2): a S-curve order that traverses rows in alternating directions, and a Hilbert

space-filling curve order that generates nearby pixels in the image consecutively. In all these classes, and a random class, where the sequence of latent variables is randomly sorted. Similar to the raster class, we augmented the number of possible orderings by reflecting and transposing the images, generating in total 8 different orderings per class.

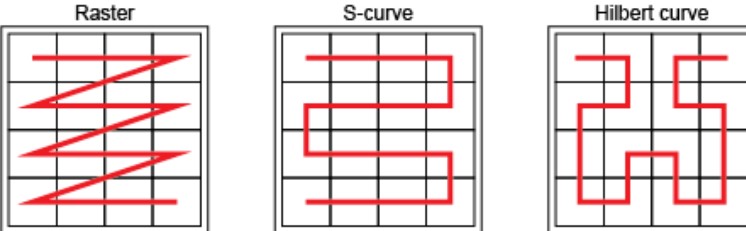

Figure A2: Different orderings used to transform the 2D discrete representation into a sequence.

In the Table A1, we can observe the performance of each class. The orderings had a performance varying from 0.843 to 0.895. We can observe that the random ordering performed with the lowest performance. This was expected since we would expect that local information of the image would help the transformer modelling of the healthy data. Since the random ordering may not include the local data in the context to predict a variable value autoregressively, this might reduce its performance as anomaly detector too. Finally, we evaluated the performance when combining all the orderings. It was observed a small gain when using an ensemble of all four classes compared to the raster class only. We opt to use the raster ordering in the main analysis to reduce the time of training and processing.

Table A1: Performance of different classes of ordering and the ensemble with all classes.

| Method | $\lceil DICE \rceil$ |
|---|---|
| 8 different raster orderings | 0.895 |
| 8 different S-curve orderings | 0.883 |
| 8 different Hilbert curve orderings | 0.890 |
| 8 different random orderings | 0.843 |
| 32 different orderings | **0.899** |

**Same ordering but different random seed**  The ensemble of 8 transformers gave a boost in performance in the segmentation comparing with a single transformer. To verify if it was due to the variability of the generative models with different orderings instead of using an ensemble with more models, we trained 8 models using the same raster ordering but with different random seed. We can observe a drop in performance when using an ensemble of transformers using the same ordering but different random seeds, from 0.895 to 0.826.

**Anomalies intensity**  We also evaluated the influence of the synthetic anomalies' intensity and texture. For this, we varied the intensity of the sprites in the image from 0 to 1 and

measured the segmentation performance (best achievable DICE-score). We also performed this approach with the addition of an additive Gaussian noise with standard deviation of 0.2. From Figure A3, we can observe that our transformer-based method is more robust to the change in intensity with a narrow drop in performance when the anomaly intensity is closer to the tissue mean values.

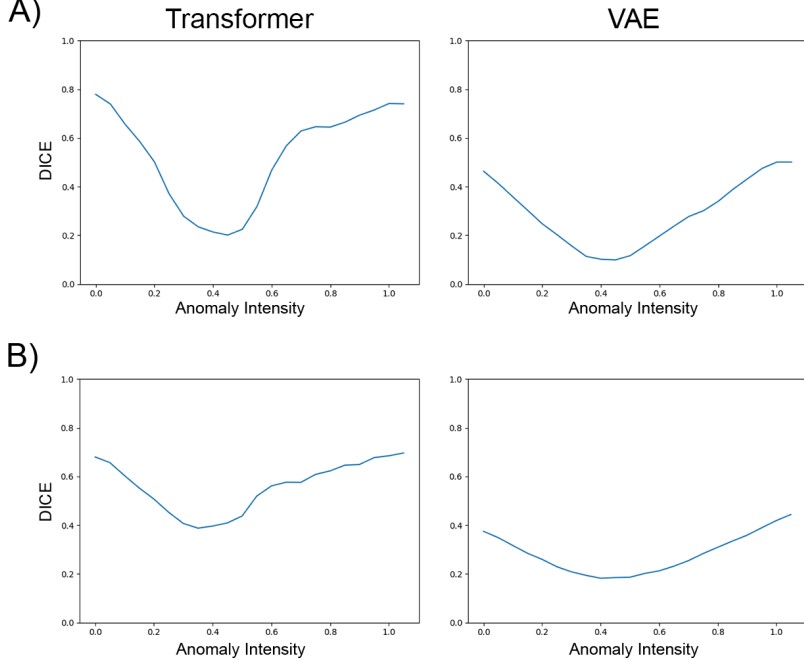

Figure A3: Analysis of the intensity of the anomalies A) with and B) without the additive noise.

## A.2. Experiment #2 – Image-wise Anomaly Detection on Synthetic Data

**Dataset**   In this experiment, we used the same training set from the Experiment #1. For evaluation, we used 1,000 images from the HeadCT class as the in-distribution test set, the 100 HeadCT images contaminated by sprites anomalies as the near out-of-distribution set (near OOD), and 1,000 images of each other classes from the MedNIST dataset (AbdomenCT, BreastMRI, CXR, ChestCT, and Hand) as the far out-of-distribution set (far OOD) (Figure A4). To train our general purpose VQ-VAE, we added 8,000 images from each other classes to our training set and 1,000 images to our validation set.

**Models and Training Settings**   To train the VQ-VAE with general purpose and its transformers, we used the same architecture, and the same training settings from the models from Experiment #1.

**Performance of Experiment #1 models on Anomaly Detection**   Besides the VAE, in Table A2 we also present the performance of the other autoencoder models from Experiment #1 to perform the anomaly detection task.

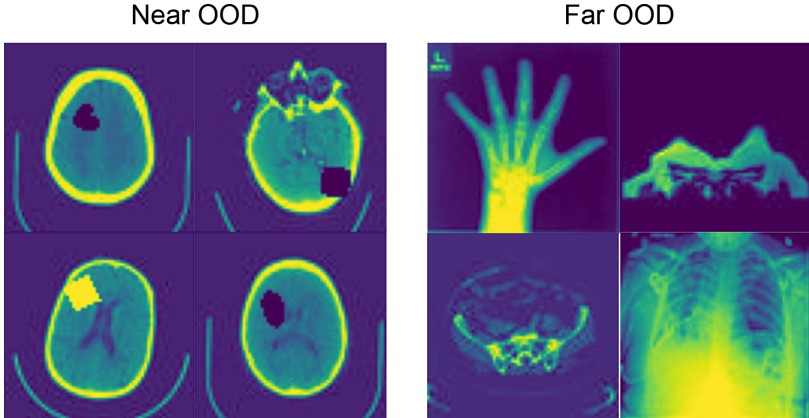

Figure A4: Examples from the near out-of-distribution set (near OOD) and far out-of-distribution set (far OOD).

Table A2: Performance of the methods on image-wise anomaly detection.

| | AUCROC | AUPRC In | AUPRC Out | FPR95 | FPR99 | FPR999 |
|---|---|---|---|---|---|---|
| vs. far OOD classes | | | | | | |
| AE (Dense) (Baur et al., 2020a) | **0.351** | **0.875** | **0.066** | 0.969 | **0.987** | **0.987** |
| AE (Spatial) (Baur et al., 2020a) | 0.337 | 0.874 | 0.063 | **0.959** | 0.998 | 0.998 |
| VAE (Dense) (Baur et al., 2020a) | 0.298 | 0.855 | 0.060 | 0.986 | 0.996 | 0.996 |
| VQ-VAE (reconstruction-based; van den Oord et al., 2017) | 0.241 | 0.834 | 0.056 | 0.987 | 0.996 | 0.996 |
| vs. near OOD classes | | | | | | |
| AE (Dense) (Baur et al., 2020a) | 0.106 | 0.093 | 0.669 | **1.000** | **1.000** | **1.000** |
| AE (Spatial) (Baur et al., 2020a) | **0.215** | **0.103** | **0.721** | **1.000** | **1.000** | **1.000** |
| VAE (Dense) (Baur et al., 2020a) | 0.111 | 0.094 | 0.672 | **1.000** | **1.000** | **1.000** |
| VQ-VAE (reconstruction-based; van den Oord et al., 2017) | 0.024 | 0.089 | 0.645 | **1.000** | **1.000** | **1.000** |

**Impact of the general VQ-VAE on anomaly segmentation** The VQ-VAE with general purpose reduce the impact of the encoder to the analysis, working mainly as a compression mechanism. In this analysis, we evaluate the performance of this models when performing anomaly segmentation. Using the general purpose VQ-VAE, we obtained a DICE score of 0.886, just a small decrease compared to the models trained only on HeadCT data.

### A.3. Experiment #3 − Anomaly Segmentation on Real Data

**MRI Datasets** In our experiment, we used three datasets: the UK Biobank (UKB) dataset (Sudlow et al., 2015), the White Matter Hyperintensities Segmentation Challenge (WMH) dataset (Kuijf et al., 2019), the Multimodal Brain Tumor Image Segmentation Benchmark (BRATS) dataset (Bakas et al., 2017, 2018; Menze et al., 2014), and the Multiple Sclerosis dataset from the University Hospital of Ljubljana (MSLUB) dataset (Lesjak et al., 2018) (Figure A5).

The UKB is a study that aims to follow the health and well-being of 500,000 volunteer participants across the United Kingdom. From these participants, a subsample was chosen to collect multimodal imaging, including structural neuroimaging. Here, we used an early release of the project's data comprising of 33,318 HC participants. More details about the

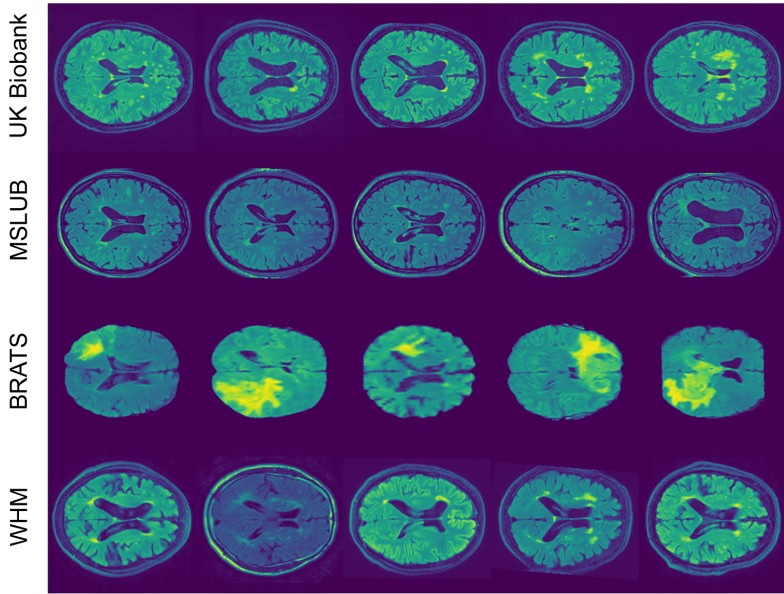

Figure A5: Examples from the test set of the Experiment #3.

dataset and imaging acquisition can be found elsewhere (Alfaro-Almagro et al., 2018; Elliott and Peakman, 2008; Miller et al., 2016). The UK Biobank dataset has available a mask for hyperintensities white matter lesions obtained using BIANCA (Griffanti et al., 2016; Jenkinson et al., 2012). We selected the 15k subjects with the lowest lesion volume to train our VQ-VAE model.

The BRATS challenge is an initiative that aims to evaluate methods for the segmentation of brain tumours by providing a 3D MRI dataset with ground truth tumour segmentation annotated by expert board-certified neuroradiologists (Bakas et al., 2017, 2018; Menze et al., 2014). Our study used the 2018 version of the dataset composed by the MR scans of 420 patients with glioblastoma or lower grade glioma. The images were acquired with different clinical protocols and various scanners from multiple (n=19) institutions. Note, the available images from the BRATS dataset were already skull stripped.

The WHM dataset is an initiative to directly compare automated WMH segmentation techniques (Kuijf et al., 2019). The dataset was acquired from five different scanners from three different vendors in three different hospitals in the Netherlands and Singapore. It is composed by 60 subjects where the WMH were manually segmented according to the STandards for ReportIng Vascular changes on nEuroimaging (STRIVE) (Wardlaw et al., 2013).

The MSLUB dataset is a publicly available dataset for validation of lesion segmentation methods. The dataset consists of 30 images from multiple sclerosis patients that were acquired using conventional MR imaging sequences. For each case, a reference lesion segmentation was created by three independent raters and merged into a consensus. This way, we have access to a precise and reliable target to evaluate segmentation methods. Full

description regarding data acquisition and imaging protocol can be found at Lesjak et al. (2018).

In our study, we used the FLAIR images from these three datasets to evaluate our models. For each of these FLAIR images, the UKB and WMH datasets had a white matter hyperintensity segmentation map, the BRATS dataset had a tumour segmentation map, and the MSLUB dataset had a white-matter lesion segmentation map. We also used T1w and brain mask to perform the MRI pre-processing.

**MRI Pre-processing**  We pre-process our images to be normalized in a common space. For this reason, all scans and lesion masks were registered to MNI space using rigid + affine transformation. This registration was performed using the Advanced Normalisations Tools (ANTs - version 2.3.4) (Avants et al., 2011). Since our anomaly segmentation method relies on a training set composed by a population with a low occurrence of lesions and anomalies, we tried to minimize the occurrence of lesions on the transformers' training set. For this reason, after the traditional MRI pre-processing, we used the NiftySeg package (version 1.0) (Prados et al., 2016) to mitigate the influence of the lesions present our training set. Using the seg_FillLesions function and the lesion maps supplied by the UKB dataset, we in-painted the few white matter hyperintensities present in the FLAIR images using a non-local lesion filling strategy based on a patch based in-painting technique for image completion. Since the VQ-VAE performs mainly a dimensionality reduction in our method, it was trained using the normalized dataset without the NiftySeg in-painting. We believe that the presence of the lesions in the VQ-VAE training set is important to avoid that the autoencoder method avoids performing the correction on its encoding part. If the encoder corrects the code by itself, the transformer would not be able to detect the presence of a lesion. This missing detection would result in a resampling mask that filters out the encoder correction. On Experiment #2, we show that this approach does not prejudice the performance of the segmentation. Finally, we selected 4 axials slices (z=89, 90, 91, 92) per FLAIR image and, we center cropped these slices to have the dimensions of 224 x 224 pixels.

**Models**  Our VQ-VAE models had a similar architecture from the Experiment #1 but with three residual 3×3 blocks (instead 2). All the convolution layers had 256 hidden units. The inputted images had 224x224 pixels which result in a latent representation of 28x28 latent variables. For this experiment, we used a codebook with 32 different codes. Our performers corresponded to the transformer's decoder structure with 16 layers with an embedding size of 256. The embedding, feed-forward and attention dropout had a probability of 0.3.

**Training settings**  To train the VQ-VAE models, we use the ADAM optimiser with learning rate 1e-3, an exponential learning rate decay with gamma of 0.99995, and we trained over 500 epochs with batch-size 256. We train the codebook similar to Experiment #1. To train the performers, we used the ADAM optimiser with learning rate 1e-3, an exponential learning rate decay with gamma of 0.99992, and we trained over 150 epochs with batch-size 128. We used data augmentation to increase the number of training images. We randomly performed small translation transformations as well as random intensity shift and random adjusts in contrast.

**State-of-the-art Models**  In this experiment, we used the network architecture from Baur et al. (2020a) for the autoencoder only-based approaches. Even with our bigger input

images, we opted to use the same instead a version with an extra downsampling step to avoid a model structure with a bottleneck with a much smaller resolution than our proposed method. To train these models, we used the ADAM optimiser with learning rate 1e-3, an exponential learning rate decay with gamma of 0.99995, and we trained over 1,500 epochs with batch-size 256. Similar to Baur et al. (2020a), we used the held-out validation set to select the models to be assessed.

**Impact of Mitigating Lesions on the Training set**  In our pre-processing, we in-painted the white matter hyperintensity of the training set using the NiftySeg package, as to simulate completely lesion-free data. Here, we compare the performance of our method without this step. By skipping this pre-processing step and training the transformers on the new dataset, we can observe a drop in performance, from 0.232 to 0.051 in the UKB dataset, from 0.378 to 0.264 in the MSLUB dataset, from 0.759 to 0.677 in the BRATS dataset, and from 0.429 to 0.349 in the WMH dataset. We believe that the highly expressive transformers can learn the few white matter hyperintensities present in the dataset and associate a higher probability of occurrence, reducing the performance in detection.

**Performance of different stages of the method**  Like Experiment #1, we evaluated the performance of each step of our method. From Table A3, we can observe that each step presents an incremental improvement.

Table A3: Performance of each processing step of our transformer-based approach in the Experiment #3

| UKB Dataset | $\lceil DICE \rceil$ |
|---|---|
| VQ-VAE (reconstruction-based; van den Oord et al., 2017) | 0.028 |
| VQ-VAE + Transformer (Ours) | 0.079 |
| VQ-VAE + Transformer + Masked Residuals (Ours) | 0.104 |
| VQ-VAE + Transformer + Masked Residuals + Different Orderings (Ours) | **0.232** |
| **MSLUB Dataset** | |
| VQ-VAE (reconstruction-based; van den Oord et al., 2017) | 0.040 |
| VQ-VAE + Transformer (Ours) | 0.097 |
| VQ-VAE + Transformer + Masked Residuals (Ours) | 0.234 |
| VQ-VAE + Transformer + Masked Residuals + Different Orderings (Ours) | **0.378** |
| **BRATS Dataset** | |
| VQ-VAE (reconstruction-based; van den Oord et al., 2017) | 0.331 |
| VQ-VAE + Transformer (Ours) | 0.431 |
| VQ-VAE + Transformer + Masked Residuals (Ours) | 0.476 |
| VQ-VAE + Transformer + Masked Residuals + Different Orderings (Ours) | **0.759** |
| **WMH Dataset** | |
| VQ-VAE (reconstruction-based; van den Oord et al., 2017) | 0.100 |
| VQ-VAE + Transformer (Ours) | 0.205 |
| VQ-VAE + Transformer + Masked Residuals (Ours) | 0.269 |
| VQ-VAE + Transformer + Masked Residuals + Different Orderings (Ours) | **0.429** |

**Post-processing Impact**  Similar to Baur et al. (2020a), we verified the performance of the methods using the prior knowledge that multiple sclerosis lesions would appear as positive residuals as these lesions appear as hyper-intense in FLAIR images. We assumed the same for the WMH lesions. By using only the positive values of the residuals as a post-processing step, we got a gain on performance on both the autoencoders only based methods and our approach (Table A4).

Table A4: Performance of each method using post-processing step.

| UKB Dataset | $\lceil DICE \rceil$ |
|---|---|
| AE (Dense) (Baur et al., 2020a) | 0.079 |
| AE (Spatial) (Baur et al., 2020a) | 0.054 |
| VAE (Dense) (Baur et al., 2020a) | 0.071 |
| VQ-VAE (reconstruction-based; van den Oord et al., 2017) | 0.056 |
| VQ-VAE + Transformer + Masked Residuals + Different Orderings (Ours) | **0.297** |
| **MSLUB Dataset** | |
| AE (Dense) (Baur et al., 2020a) | 0.106 |
| AE (Spatial) (Baur et al., 2020a) | 0.067 |
| VAE (Dense) (Baur et al., 2020a) | 0.106 |
| VQ-VAE (reconstruction-based; van den Oord et al., 2017) | 0.077 |
| VQ-VAE + Transformer + Masked Residuals + Different Orderings (Ours) | **0.465** |
| **WMH Dataset** | |
| AE (Dense) (Baur et al., 2020a) | 0.166 |
| AE (Spatial) (Baur et al., 2020a) | 0.151 |
| VAE (Dense) (Baur et al., 2020a) | 0.161 |
| VQ-VAE (reconstruction-based; van den Oord et al., 2017) | 0.143 |
| VQ-VAE + Transformer + Masked Residuals + Different Orderings (Ours) | **0.441** |

## Appendix B. More Residuals maps from Experiment #1

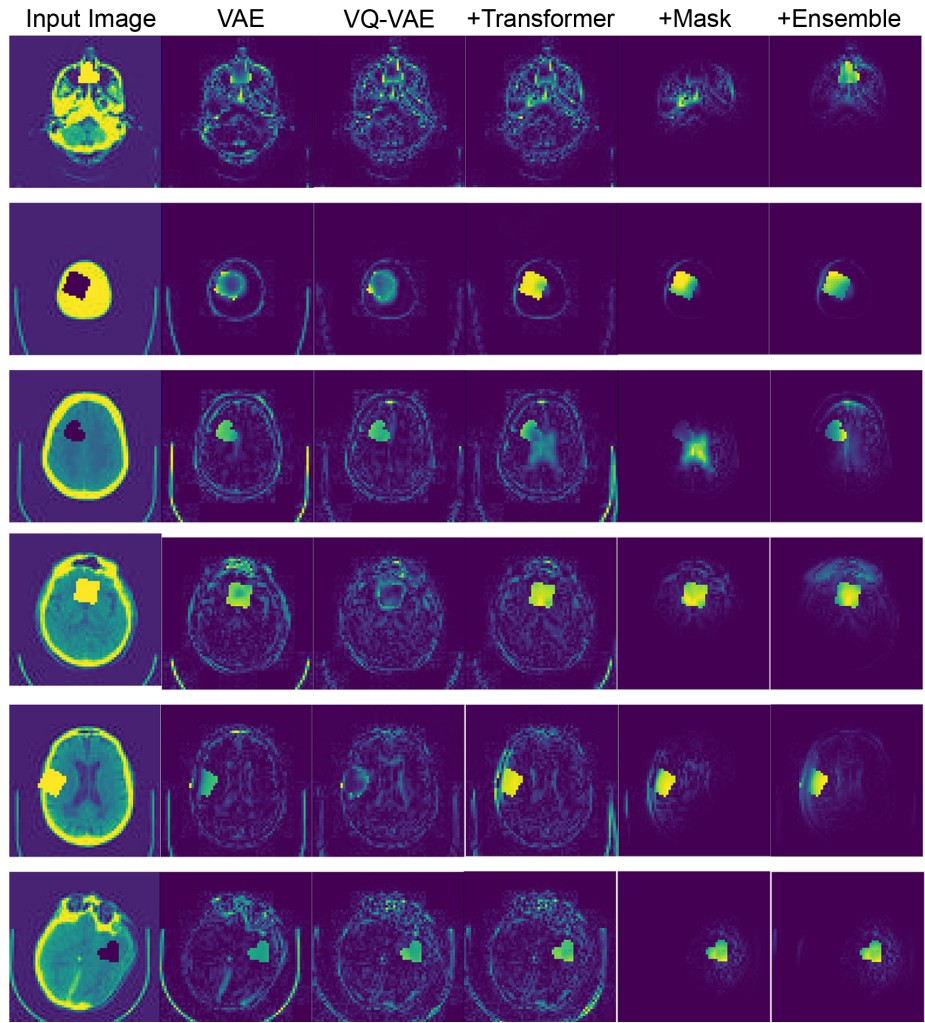

Figure A6: More examples of residual maps on the synthetic dataset from the variational autoencoder and different steps of our approach.

## Appendix C. More Residuals maps from Experiment #3

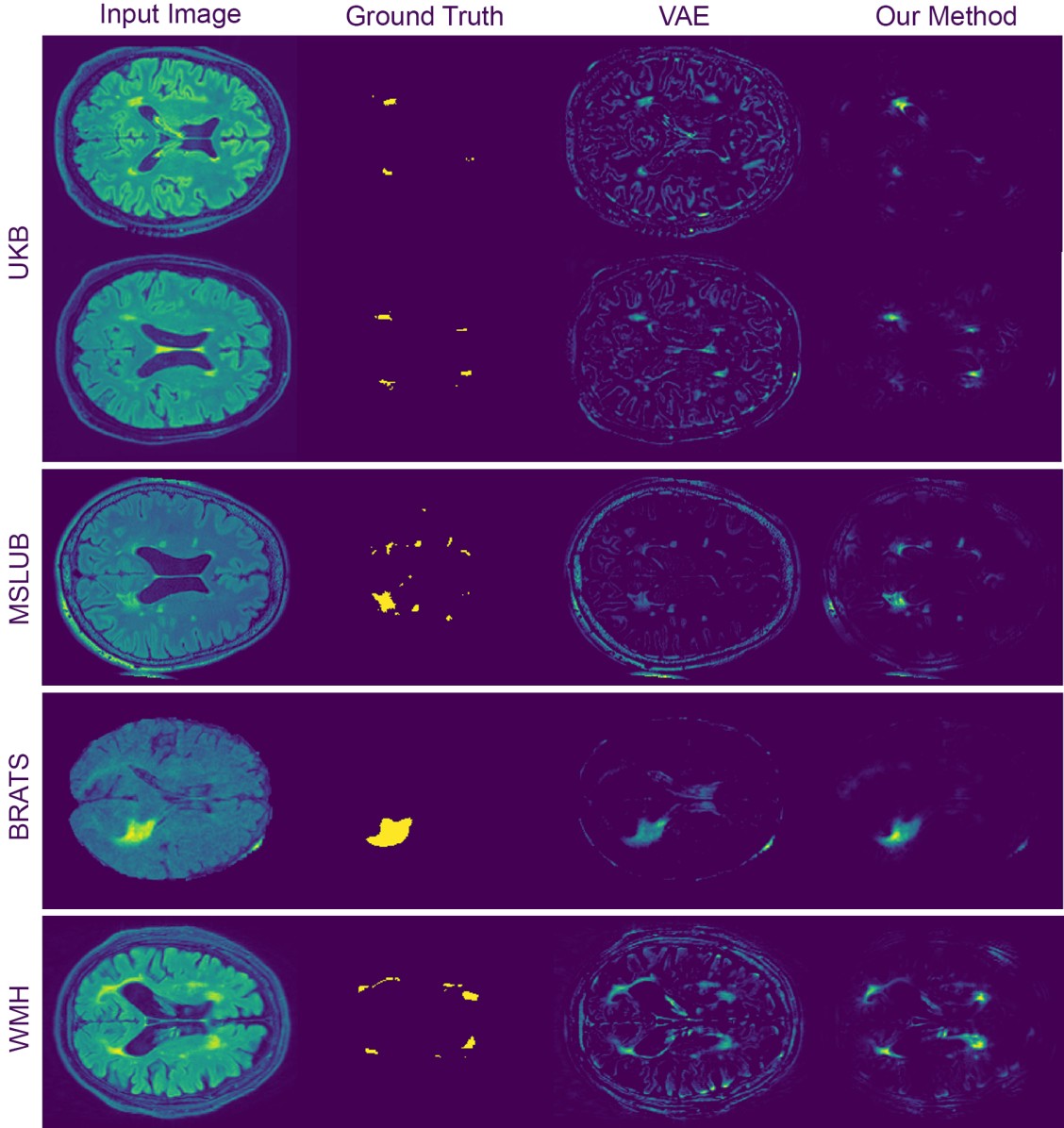

Figure A7: More examples of residual maps on the real lesions from the variational autoencoder and our method.

