# OpenReview forum: "Unsupervised Brain Anomaly Detection and Segmentation with Transformers"
_MIDL.io/2021/Conference — MIDL 2021_

### Official Review · AnonReviewer3 · 2021-03-08

**Confidence:** 4
**Preliminary Rating:** 4
**Recommendation:** Best Paper Award, Oral
**Final Rating:** 4

**Summary:**

This paper proposes a new approach using transformers with vector quantized variational autoencoders (VQ-VAE) for unsupervised anomaly segmentation and detection in brain MRI. The VQ-VAE is used to efficiently encode the images, whose latent encoding is then analyzed via the transformer to learn the distribution of encodings for normal data by training in an autoregressive fashion. Detection of anomalies in new images can then be performed by encoding the input image, finding the portions of the latent code that are low probability of being normal, replacing these values with generated values sampled from the transformer model, and then reconstructing a "normal" version of the input image and comparing to the original image.  Their method was tested on both a synthetic data created using the MedNIST dataset and on brain data with real abnormalities, training on the UK Biobank dataset and testing on 4 different datasets with white matter hyperintensities, in which the proposed approach demonstrated substantial increases in accuracy compared to recent methods using AEs or VAEs.

**Strengths:**

1. The proposed approach utilizes the power of the transformer model, which is now state of the art in language modeling, for static neuroimaging data. Transformers have seen limited adoption so far in medical imaging domain and given the success of these models in other domains, their application in medical imaging should be encouraged.

2. The authors circumvent the computational difficulty with using transformer models by applying the VQ-VAE first to reduce the input dimension and then also adopting performers, a new model with efficient implementation of the attention mechanism.

3. The experimental validation is quite extensive, with 3 different experiments (2 on synthetic data, 1 on real data), using very large datasets (~10000 images for synthetic, training on 15000 from UK biobank for real data experiments), and included evaluation on multiple datasets with actual brain abnormalities. For each of these experiments, the authors reported substantial increases in anomaly detection/segmentation performance compared to recently published baseline methods.

4. The paper is very well written and easy to follow.

**Weaknesses:**

1. The evaluation of segmentation is measured using "highest-achievable" dice score, where a search through residual thresholds was done and the best possible result was reported. I understand this is done for all the methods and thus is fair in that sense, but it would be nice to know what the accuracy would be like under normal use cases. Given the large amount of data, I'm wondering why a validation set was not used to choose a threshold? Also, maybe could have used AUC of segmentation like in the anomaly detection experiment so that it is a threshold-free way of comparing the models.

2. I am curious as to whether the authors can explain why the different orderings may have an effect on the result, when the advantage of the transformer is that it is supposed to be able to learn complex dependencies across any range of sequences?

3. This is not a weakness per se, but a thought. The anomaly detection is done in latent space, using the likelihood of the latent encoding, while the anomaly segmentation is done using comparison in original image space. I'm wondering if the detection of anomalous regions could be done in the latent space first, then transformed back to image space, or whether the low dimensional encoding is prohibitive and loses spatial resolution.

**Deanonymize Review:**

no

**Detailed Comments:**

While the paper is really well written, a proofread to fix some missed typos would polish the paper.



**Final Rating Justification:**

The authors addressed/answered my questions (and also responded well in their rebuttal to other reviewers) and kindly added my suggested AUC metric to experimental results for another view of model performance, which further strengthens their results. Therefore, I keep my original rating of strong accept.

**Justification Of The Preliminary Rating:**

The authors present an exciting application of transformers to the medical imaging domain. Furthermore, through extensive validation on large datasets, they showed substantial improvements in brain anomaly segmentation and detection.

**Paper Type:**

both

**Special Issue:**

yes

---

> ### Author Response · Authors · 2021-03-17
> **Response to AnonReviewer3**
>
> Thank you so much for your comments. We really appreciate the detailed summary of our paper, and we are very thankfully for your recommendations. We address your points below:
>
> ---
>
> ***1.	“Given the large amount of data, I'm wondering why a validation set was not used to choose a threshold? Also, maybe could have used AUC of segmentation like in the anomaly detection experiment so that it is a threshold-free way of comparing the models.”***
>
> That is a good point. Indeed, a subset from UK Biobank, or even from the test datasets (to avoid issues between different scanners/studies), could be used to find an optimal threshold, and then we could report the mean and standard deviation of the DICE scores across all images. Unfortunately, owing to the proximity of the MIDL deadline, we were not able to perform these analyses in time. AUPRC are now added to Tables 1 and 3 as requested.
>
> ---
>
> ***2.	“I am curious as to whether the authors can explain why the different orderings may have an effect on the result, when the advantage of the transformer is that it is supposed to be able to learn complex dependencies across any range of sequences?”***
>
> That is a great question. The autoregressive nature of our transformers means they will use the past latent variables as “context” when predicting the probability of a latent value. Anomalies will differ in their identifiability with variations in the parts of the image by which they are contextualised. For example, anomalies in the left hemisphere can be easier to identify if the model is considering the context of the homologous part of the right hemisphere (because the brain is largely symmetrical) than if it is considering the background of the image.
>
> By ordering the latent space in different ways, the various models will be forced to learn different interactions between parts of the image based on their availability in the model’s context.
>
> ---
>
> ***3.	“I'm wondering if the detection of anomalous regions could be done in the latent space first, then transformed back to image space, or whether the low dimensional encoding is prohibitive and loses spatial resolution.”***
>
> This is a great idea, and we believe that it is what our “upscaled resampling mask” is performing (Section 3.2). We observed that the anomalous regions identified in the latent space contained coarse-grained information about the location of the anomalies in the image space. As expected, it lacks precision when delineating the contours of the anomalies. However, we still make use of this spatial information from the latent space by using an upscaled version of the resampling mask as a "filtering mask" that we apply in the resulting difference between the reconstructed "normal" version of the input image and the original image. To simplify, we are obtaining the intersection between what was found in the latent space and what was found in the difference between the “normal” reconstruction and the original image.
>
> We observed that this filtering operation removed part of the false positives created by the VQ-VAE decoder's limitations in recreate sharp reconstructions in regions with high-frequency patterns.
>
> ---
>
> Again, thank you for taking the time to read our draft. We look forward to further discussion of these points.

---

### Official Review · AnonReviewer4 · 2021-03-08

**Confidence:** 5
**Preliminary Rating:** 3

**Summary:**

Paper #42
Unsupervised Bran Anomaly Detection and Segmentation with Transformers

Summary
This paper proposes to segment image anomalies by learning image normality from healthy subjects. A VAE-based approach allows to determine what diverge from normality via a resampling masks, which in turns the generation of what an expected healthy brain area should be. Evaluation is made by learning normality from the UKBiobank and testing on established labeled datasets.


**Strengths:**

Strengths
- The idea of exploiting a resampling masks appears original and shows an improvement over simple auto-encoder approaches.
- Unsupervision offers practical advantages.
- Use of ensembles for added robustness is welcome.


**Weaknesses:**

Weaknesses
- Motivation of Transformers and Performers could be strengthened. As is, more details could be useful. This may further highlight the differentiation of the proposed method with the VQ-VAE.
- Results from a vanilla VQ-VAE appears very low when compared to the proposed enhancement of the method. This should be further verified.


**Deanonymize Review:**

no

**Detailed Comments:**

Detailed Comments
- UKBiobank consists of unlabeled data, how do Dice scores appears in Table 3 for this dataset?
- Lengthy appendix, which may give an impression that important extra details and added experiments has been deferred in extra pages. The key missing info from the main article should be included in the paper, for reproducibility and consistency purposes.


**Justification Of The Preliminary Rating:**

- Recommendation for Weak Accept.
- Method offers advantages (Unsupervision, original concepts such as resampling masks).
- Evaluation on UKBiobank is good but clarification on public labels is to be provided or on reproducibility.
- Improvement on motivation.


**Paper Type:**

methodological development

**Special Issue:**

no

---

> ### Author Response · Authors · 2021-03-17
> **Response to AnonReviewer4**
>
> Thank you for your comments. We are glad to hear that you found our use of resampling masks interesting. This mechanism was crucial to reducing the incidence of false positives and – combined with the ensemble of transformers that learned to identify anomalous latent values contextualised by different parts of the latent space - added robustness that gave us substantial improvements over the other autoencoder approaches.
>
> We hope to address your concerns in the points below.
>
> ---
>
> ***1.	“Motivation of Transformers and Performers could be strengthened. As is, more details could be useful.”***
>
> An autoregressive model performing an explicit density estimation of the latent variable of the VQ-VAE model was crucial to implementing the mechanisms on which successful anomaly detection depends here. The autoregressive aspect gives us multiple views of the latent space (Section 3.3). As highlighted in the first paragraph of the introduction, we chose transformers for this aspect owing to their state-of-the-art performance across a wide variety of tasks, including autoregressive image modelling. Transformer-based approaches have consistently outperformed other autoregressive models in recent studies (e.g., convolutional-based ones, like PixelSnail) [1-5]. These considerations motivated their use here.
>
> We have elaborated on the motivation for using transformers as follows (Section 2.2).
>
> > “After training the VQ-VAE, we can learn the probability density function of the latent representation of healthy brain images using an autoregressive model. In recent studies, the transformer-based approaches have consistently outperformed other autoregressive models (Esser et al., 2020). The defining characteristic of a transformer is that it relies on attention mechanisms…”
>
> The success of transformers in other areas, such as natural language processing, has driven the optimisation of their computational requirements and performance in the context of long sequences. The advances will be essential to generalising our approach across other types of medical images, at higher resolutions and dimensionalities.
>
> [1] - Esser, Patrick, Robin Rombach, and Björn Ommer. "Taming Transformers for High-Resolution Image Synthesis." arXiv preprint arXiv:2012.09841 (2020)
>
> [2] – Parmar, Niki, et al. "Image transformer." International Conference on Machine Learning. PMLR, 2018.
>
> [3] – Ho, Jonathan, et al. "Axial attention in multidimensional transformers." arXiv preprint arXiv:1912.12180 (2019).
>
> [4] – Child, Rewon, et al. "Generating long sequences with sparse transformers." arXiv preprint arXiv:1904.10509 (2019).
>
> [5] – Chen, Mark, et al. "Generative pretraining from pixels." International Conference on Machine Learning. PMLR, 2020.
>
> ---
>
> ***2.	“Results from a vanilla VQ-VAE appears very low when compared to the proposed enhancement of the method. This should be further verified.”***
>
> The vanilla VQ-VAE operated identically to the other autoencoder approaches, quantifying anomalies by the difference between the input image and the reconstructed one. The distinctive characteristic was the use of discrete latent variables. As shown by the results, we believe that the difference between continuous or discrete latent variables does not impact on the performance of the autoencoder-based model on unsupervised anomaly detection and segmentation tasks: the critical architectural differences are downstream.
>
> We tried to make it clearer in the new version of the text by referring to it as *“VQ-VAE (reconstruction-based; van den Oord et al., 2017)”* in Tables 1, 3, A2, A3, and A4.
>
> ---
>
> ***3.	“UKBiobank consists of unlabeled data, how do Dice scores appears in Table 3 for this dataset?”***
>
> UK Biobank now includes automated segmentations of white matter hyperintensities. As mentioned in Section A.3. - MRI Datasets, each subject's scan was accompanied by a lesion mask. This lesion segmentation was automatically carried out using the discrepancies between the T1 and FLAIR images calculated by the BIANCA tool (from FMRIB Software Library - FSL). This is part of the image processing described in Alfaro-Almagro et al., 2018 and the script to obtain it is available at [https://git.fmrib.ox.ac.uk/falmagro/UK_biobank_pipeline_v_1/-/blob/master/bb_structural_pipeline/bb_BIANCA](https://git.fmrib.ox.ac.uk/falmagro/UK_biobank_pipeline_v_1/-/blob/master/bb_structural_pipeline/bb_BIANCA) .
>
> Unfortunately, due to the page limit, we could not include detailed descriptions of the MRI datasets and pre-processing in the main body of the paper without significantly shortening more important sections. For this reason, this information is located in the appendix.
>
> ---
>
> Again, thank you for taking the time to read our draft. We look forward to further discussion of these points.

---

### Official Review · ~Dewei_Hu1 · 2021-03-09

**Confidence:** 4
**Preliminary Rating:** 4
**Recommendation:** Oral
**Final Rating:** 4

**Summary:**

The key idea is first use the VQ-VAE to do dimension reduction to the original data and have it represented by a latent space. The latent vector (with dimension of $n_{z}$) at each location is discretized by a codebook ($e_{i}$, where $i=1,2,...,K$). Then the latent image ($z_{q} \in R^{h\times w}$) can be translated to a indices vector $s$ with length of $h\times w$.
The transformers is used to learn the pdf of healthy representation in latent space. Hence, the probability of being healthy for each entry if sequence $s$ can be computed ($p(s_{i})$) by transformer. A empirical threshold (0.005) is set in order to classify abnormal latent representations.  These anomaly entries are then replaced by healthy samples generated by the transformer. Then the 'healed' version of latent sequence $\hat{s}$ is decoded back to the original resolution $\hat{x}$. Finally, the residual defined by the difference between the healed image and the original input image (anomaly data) $\|x-\hat{x}\|$ is used to find the part that needs to be segmented (manual thresholding required).

**Strengths:**

Playing around the lower dimensional latent space is interesting and shows potential for further discussion. And its property makes tools designed for sequence data (transformer/LSTM) applicable for high dimensional data.
The illustration of the paper is clear and professional.

**Weaknesses:**

Personally I am curious about how the latent codebook ($e_{1},e_{2},...e_{K}$) is set up which is not mentioned in the paper.
Secondly, in the evaluation part, the real lesion segmentation dice score is poor for all baseline methods (a lot of false positive). Skull stripping and more advanced binarization methods might be able to help with the performance.

**Deanonymize Review:**

yes

**Final Rating Justification:**

The author clearly answer the question and there is no obvious weakness in the paper.

**Justification Of The Preliminary Rating:**

The author presents a unsupervised method for MRI lesion segmentation with good novelty and interpretability. The manipulation of latent space can be extended to other unsupervised segmentation problems.

**Paper Type:**

validation/application paper

**Special Issue:**

yes

---

> ### Author Response · Authors · 2021-03-17
> **Response to AnonReviewer5**
>
> Thank you so much for your comments. We appreciate the cogent summary of our study.
>
> We are grateful for your highlighting the potential of exploring the latent space, using a state-of-the-art sequence model to “heal” the latent sequence. The application of highly expressive transformers to high dimensional data made it possible to achieve excellent anomaly detection performance, and to create a novel way of exploiting spatial information in the latent space (resampling mask) to filter and improve the segmentation map.
>
> We address your queries below:
>
> ---
>
> ***1.	“Personally I am curious about how the latent codebook is set up which is not mentioned in the paper.”***
>
> Thank you for highlighting this omission in the main text. Unfortunately, due to the page limit, we decided to move the training details, such as the latent codebook training, to the appendix. In A1 - Training settings (page 13), we mention:
>
> > “We train the codebook using an exponential moving average algorithm. To stabilise the codebook's learning, in the first 100 epoch, we warm up the moving average decay from a gamma decay of 0.5, gradually increasing to a gamma decay of 0.99. This allows the codes to adapt faster to the frequent changes at the beginning of the training.”
>
> ---
>
> ***2.	“Secondly, in the evaluation part, the real lesion segmentation dice score is poor for all baseline methods (a lot of false positive). Skull stripping and more advanced binarization methods might be able to help with the performance.”***
>
> We agree that further pre- and post-processing would improve performance. In section A3 - Post-processing Impact, we showed that post-processing informed by prior knowledge lesion type improves performance across models. This observation agrees with Baur et al. (2021), who use slightly eroded brain-mask to remove prominent residuals occurring near sharp edges at brain-mask boundaries and at the cortical surface.
>
> In our study, however, we deliberately kept pre-processing to a minimum to maximise the stringency of the test and reassure us of the robustness and generality of the approach. The idea was to show that its performance was not dependent on the pre-processing performed, making it a good candidate for anomaly detection in different types of medical images or for anomalies in general. We wanted to show that our method is robust in such conditions.
>
> The use of other pre-processing and its impact could be explored in further experiments (that could also be added in the appendix, like the A3 - Post-processing Impact). However, owing to the appendix's large size, we decided not to explore these points in the paper.
>
>
> Baur, Christoph, et al. "Autoencoders for unsupervised anomaly segmentation in brain mr images: A comparative study." Medical Image Analysis (2021): 101952.
>
> ---
>
> Again, thank you for taking the time to read our draft. We look forward to further discussion of these points.

---

### Official Review · AnonReviewer2 · 2021-03-10

**Confidence:** 5
**Preliminary Rating:** 2

**Summary:**

This submission handles the unsupervised anomaly detection and segmentation problem using the combination of the vector quantized variational autoencoder (VQ-VAE) and the transformers. The proposed method was evaluated on multiple datasets and was demonstrated with improvements over AE and VAE methods.

**Strengths:**

The paper targets solving a challenging and interesting task in medical image analysis. The method was tested on multiple datasets. It shows a decent improvement over AE and VAE methods with a big margin.

**Weaknesses:**

1) The novelty of this submission is limited. It's an application of the existing works VQ-VAE and transformers.

2) The paper didn't mention or compare to the state-of-the-art work in unsupervised anomaly detection, i.e., the AnoGAN and its family like f-AnoGAN, VAE-ANOGAN.

Thomas Schlegl et al., Unsupervised Anomaly Detection with Generative Adversarial Networks to Guide Marker Discovery, IPMI 2017.


**Deanonymize Review:**

no

**Justification Of The Preliminary Rating:**

Lacking novelty in the proposed method and missing an important reference in the literature are the two main issues of this submission.  Both literature review and experiments need extra work, the paper is not ready for publishing.

**Paper Type:**

validation/application paper

**Special Issue:**

no

---

> ### Author Response · Authors · 2021-03-17
> **Response to AnonReviewer2**
>
> Thank you for your thoughts, AnonReviewer2. We are glad that you agree unsupervised anomaly detection and segmentation are both challenging and interesting, and appreciate that you think that our method shows decent improvement on the current state of the art. We hope to address your concerns below.
>
> ---
>
> ***1.	“The novelty of this submission is limited. It's an application of the existing works VQ-VAE and transformers.”***
>
> Much of machine learning is architecturally combinatorial: were the previous use of individual architectural elements grounds for dismissing a paper, very few would ever be published. It is true that the combination of quantized variational autoencoders with transformers is intuitively suited to the task—as we argue in the text—but realising it in practice is far from trivial, which may explain why no one has attempted it before. The computational burden (both processing and memory) and the complexities of training in the setting of long-range interactions require novel solutions not provided elsewhere. We are the first to succeed in using this architecture to perform unsupervised anomaly detection and anomaly segmentation on medical images.
>
> Moreover, as highlighted by the other reviewers, we introduce a set of other technological innovations of relevance to the broader anomaly detection literature. One is the use of resampling masks (Section 3.2) to mitigate the impact of false positives to which autoencoder-based methods are prone owing to imperfect reconstruction of high frequencies. Another is the use of different reordering of the latent embeddings within a thereby optimised ensemble method (Section 3.3). This step showed marked superiority over conventional ensembles (Section A1 - Same ordering but different random seeds).
>
> ---
>
> ***2.	“The paper didn't mention or compare to the state-of-the-art work in unsupervised anomaly detection, i.e., the AnoGAN and its family like f-AnoGAN, VAE-ANOGAN.”***
>
> Thank you for the suggestion: we now cite this paper in the introduction. The reason for focusing the comparison on a relatively small number of methods – AE, AE dense, and VAE – is because these models have been widely adopted as baselines by other unsupervised anomaly segmentation studies, and are closely illustrative of the limits of currently achievable performance. Note that in Baur et al. (2021), a recently published peer-reviewed study, an extensive comparison of different unsupervised methods yielded the following theoretically best possible DICE-score for the f-AnoGAN and the VAE:
>
> > MS Dataset: VAE=0.469 ; f-AnoGAN=0.489 (Table 3)
>
> > GB Dataset: VAE=0.441 ; f-AnoGAN=0.447 (Table 4)
>
> > MSLUB Dataset: VAE=0.323 ; f-AnoGAN=0.283 (Table 5)
>
> > MSSEG2015 Dataset: VAE=0.257 ; f-AnoGAN=0.342 (Table 6)
>
> Although f-AnoGAN was generally superior, the margins were slim. Moreover, the f-AnoGAN approach is heavily dependent on post-processing steps, as shown in Table 8 (DICE: VAE=[0.051, 0.188]; f-AnoGAN=[0.088, 0.137]) that arguably obfuscate the comparison of model performance and potentially limit generalisability. For these reasons, and pressures on the available computational resource, we did not include an f-AnoGAN model and have not been able to add it to our results in the rebuttal period.
>
> Baur, Christoph, et al. "Autoencoders for unsupervised anomaly segmentation in brain mr images: A comparative study." Medical Image Analysis (2021): 101952.
>
> ---
>
> Again, thank you for taking the time to read our draft. We look forward to continuing the discussion of these points.

---

### Meta-Review · Area_Chair1 · 2021-03-28

**Recommendation:** Accept (Oral & Special Issue Candidate)

**Metareview:**

I share the positive views of the reviewers about this paper. I was especially pleased to see the improvements made to the paper during the rebuttal phase such as the addition of the AUC experiment.

**Paper Type:**

both

---

### Decision · Program_Chairs · 2021-03-31

**Decision:**

Accept

**Comment:**

Congratulations your paper has been selected as a long oral.